# Therapist-Guided Versus Self-Guided Forest Immersion: Comparative Efficacy on Short-Term Mental Health and Economic Value

**DOI:** 10.3390/bs15121618

**Published:** 2025-11-24

**Authors:** Rosa Rivieccio, Francesco Meneguzzo, Giovanni Margheritini, Tania Re, Ubaldo Riccucci, Federica Zabini

**Affiliations:** 1Centro di Ricerca Politiche e Bioeconomia, Consiglio per la Ricerca in Agricoltura e l’Analisi dell’Economia Agraria, I-00178 Roma, Italy; rosa.rivieccio@crea.gov.it; 2Institute of Bioeconomy, National Research Council, 10 Via Madonna del Piano, I-50019 Sesto Fiorentino, Italy; federica.zabini@cnr.it; 3Central Scientific Committee, Italian Alpine Club, I-20124 Milano, Italy; margheritinig@gmail.com; 4Group Anthropology of Health—Biosphere and Healing Systems, University of Genoa, I-16128 Genoa, Italy; tania.re77@gmail.com; 5Simple Departmental Operating Unit Anesthesia and Resuscitation, Cecina Hospital, I-57023 Cecina, Italy; ubaldo.riccucci@uslnordovest.toscana.it

**Keywords:** forest therapy, forest bathing, nature-based interventions, therapist-guided, self-guided, anxiety, self-esteem, total mood disturbance, quality-adjusted life years (QALY), economic evaluation

## Abstract

Forest therapy, guided by clinical professionals (psychologists or psychotherapists), is increasingly recognized as a preventive and complementary health practice with evidence-based therapeutic potential; however, the specific contribution of therapist guidance compared to self-guided immersion remains unclear. This retrospective study evaluated the short-term mental health outcomes of therapist-guided (TG) compared to self-guided (SG) forest immersion, based on the validated State–Trait Anxiety Inventory and Profile of Mood States questionnaires. Data were collected from 282 adults participating in eight paired TG–SG sessions conducted at the same forest sites across Italy. The results showed that TG sessions were associated with greater improvements in state anxiety, self-esteem, and total mood disturbance, with statistically significant effects in most cases. Therapist-led guidance also occasionally reduced interindividual variability, suggesting enhanced emotional regulation. An illustrative economic assessment, based on standardized psychometric improvements translated into quality-adjusted life years (QALYs), indicated that the TG sessions yielded approximately 1.7 times the annual per-person economic value of the SG sessions, outweighing the associated therapist-related costs. These findings suggest that TG forest therapy interventions deliver significant and economically quantifiable added value compared to SG experiences, supporting their inclusion in preventive health and mental well-being programs and justifying further longitudinal and cost-effectiveness investigations.

## 1. Introduction

Forest therapy and forest bathing (Shinrin-yoku) are structured practices of mindful immersion in woodland environments that have gained international attention as preventive and complementary health strategies. They were established officially in 1982 by Japan’s Ministry of Agriculture, Forestry, and Fisheries to promote public health in response to rising levels of stress, anxiety, and illnesses linked to urbanization and the phenomenon of karoshi (death from overwork) ([61]). Forest therapy and forest bathing are now the subject of extensive research globally, including in European countries, with a growing body of scientific literature linking forest exposure to effective psychological and physiological benefits ([9]; [33]). Reported effects include reductions in stress and anxiety, enhanced affective well-being, and improvements in cardiovascular, immune, and respiratory functions ([26]; [40]; [41]).

Mechanisms appear to be both psychological and biological. Research suggests that forest settings facilitate attentional restoration and emotion regulation, while exposure to natural volatiles such as plant-emitted monoterpenes may exert direct anxiolytic effects ([16]). These effects have previously been confirmed, along with significant effects on the immune and endocrine systems and cardiovascular health, in experiments conducted in both controlled and forest environments ([26]). In particular, clinical studies highlighted that forest immersion can provide supportive benefits for chronic conditions such as pain, chronic obstructive pulmonary disease, and fibromyalgia ([27]; [50]; [54]).

Forest bathing is a personal health practice devoid of clinical aims, whose degree of structuring can vary from free forest walking to mindful forest immersion, sometimes along signed forest trails, to experiences guided by non-clinical personnel. Recently, research in the field of mental health suggested that nature-based walking interventions may be used to relieve at least one’s negative mood, stress, and anxiety; however, to enhance the treatment efficacy, these interventions should be combined with formal modes of psychotherapy ([29]). This leads to forest therapy, as a therapeutic intervention guided by a clinical professional, in particular a psychologist or a psychotherapist, in the following, referred to as therapist. To be substantiated, forest therapy requires an accurate identification of the scope along with the significance and size of the effects, the latter in turn conditioned on the characteristics of the intervention (environment and activities) and personal traits, including gender, age and possible active illness ([7]).

Although some reviews caution against overstating its efficacy, forest therapy is increasingly recognized as a low-risk, accessible intervention that complements conventional care and resonates with the One Health framework linking human well-being and ecosystem health ([37]). These insights highlight its relevance to contemporary health research and practice.

The question whether therapist-guided experiences actually provide an added benefit to participants’ mood and psychological symptoms compared to self-guided forest immersion experiences remains partially unanswered ([6]). In other words, does therapist-guided (TG) forest therapy really convey additional benefits to short-term mental health compared to self-guided (SG) forest bathing? To the best of the authors’ knowledge, existing studies addressing this question have predominantly focused on university or graduate student populations within campuses located in East Asian countries.

With 37 undergraduate students randomly assigned to a guided forest therapy program or self-guided forest immersion experience in a university campus in South Korea, based on free personal essays, Kim and Shin found that the self-guided experience promoted self-reflection, while guided forest therapy delivered positive emotional changes and promoted social bonds ([24]). Later, the same authors experimented with a randomized 3 × 3 crossover study, where 23 university students were randomly assigned to a self-guided forest therapy program, a guided forest-therapy program, or routine activities. They found that self-guided and guided forest-therapy programs significantly improved participants’ negative emotions and natural connectedness compared to routine activities; however, for self-esteem, only a guided forest therapy program delivered a significant improvement compared to daily routine activities ([51]).

In a similar study, Shin et al. found that self-guided and guided forest-healing programs significantly improved subjects’ mood states and anxiety symptoms compared to routine activities. Participating in a forest healing program with guides and participating in a self-guided forest healing program both provided psychological benefits for subjects, showing that self-guided programs can be effectively combined with forest healing ([52]).

In a recent large-scale experiment involving approximately 250 university students, randomly assigned to guided forest therapy or self-guided forest walking on a university campus in China, Guo et al. found no significant differences in effectiveness based on self-reported psychological and physiological measures (blood pressure and heart rate) ([21]).

This study aimed to assess the short-term effectiveness for mental health of TG forest therapy activities compared to the same activities performed through SG experiences, where the only difference was the verbal communication of the instructions (TG sessions) or the reading and interpretation of written signs placed along the forest trails (SG sessions). Besides this original experimental design, the novelty also arises from the evaluation, for the first time, across multiple sites and multiple adult age groups.

Moreover, this study aimed at providing an illustrative assessment of the absolute and comparative economic value of TG and SG programs, based on the existing evidence of the persistence of beneficial effects of forest therapy interventions and using an exploratory valuation based on quality-adjusted life years (QALYs), which is widely adopted and specifically designed for internal budget allocations within healthcare portfolios ([4]).

## 2. Materials and Methods

### 2.1. Design and Participants

Experimental sessions were conducted between September 2021 and June 2025 in 8 sites, one of which was located in an urban park. This was part of a broader experimental campaign involving tens of sites across Italy, whose data, collected from September 2021 to October 2022, were used in a previous study ([16]). Table 1 shows the geographical, vegetational, and weather data of study sites; temperature levels were measured using hand-held thermometers and reported as the average value during the TG and SG sessions, while cloudiness was visually estimated by an expert meteorologist. Table 2 shows participant data of the TG and SG sessions. Figure 1 shows the location of the eight sites considered in this study. All geospatial data were referenced to the WGS 84 geographic coordinate system (EPSG:4326). Cartographic representation of the study area and sampling sites was performed using QGIS (version 3.28 ‘Firenze’, https://qgis.org/, accessed on 6 October 2025).

An overall population of 282 adults, aged between 18 and over 70, took part in this study. For each of the eight sites, two forest immersion sessions were carried out along the same trail and starting no more than 20 min apart to remove possible systematic biases. For each site, the first group participated in a TG session, where the therapist verbally communicated the instructions, whereas the second group engaged in a SG session, having to read and interpret the same instructions displayed on signs placed along the path. The SG group was led either by one of the authors of this study or by a hiking guide, who also marked the end time of the stops at any sign; thus, the only difference compared to the TG sessions consisted in reading and interpreting the instructions written on the signs. Because participants were not randomly assigned to TG vs. SG sessions, the design is quasi-experimental and susceptible to self-selection bias.

Each session lasted in total approximately 3.0 ± 0.3 h, including the filling out of questionnaires before and after the session, while the walks lasted about 2.3 h. Data were collected immediately before and after each TG or SG session, following a pre–post study without a follow-up period.

Only adults who were able to independently complete a questionnaire written in Italian and explicitly willing to participate in forest therapy sessions after filling out their informed consent forms were included in this study. Exclusion criteria were being underage (<18 years), having any acute illnesses requiring medications beyond those usually taken daily for chronic diseases conditions, and any major physical or mental health problems incompatible with outdoor walking for a few hours. Each participant was allowed to attend only one session, to prevent any possible carry-over effect.

All participants were treated in accordance with the ethical guidelines for research provided by the Declaration of Helsinki and its subsequent revisions, and this study was approved by the local ethics committee.

The forest immersion program, both in TG and SG sessions, consisted of simple instructions to the participants to focus their attention on the external environment. The protocol was designed to provide reproducible instructions to all the therapists involved, in order to minimize any interference related to the therapists’ personal styles and ensure homogeneous experimental conditions ([16]). The overall duration of each session was approximately 3 h, including short, slow walks, interspersed with five stops focusing on senses (sight, hearing, touch, smelling, and a final multi-sensorial stop), and a final walk, before filling out the psychometric self-assessment questionnaires after the forest therapy experience. The physical intensity of the sessions was kept low to avoid potential biases due to adrenergic hyperactivation. Each session was conducted in the morning, starting around 10:00 a.m. and ending around 01:00 p.m., local time (corresponding to GMT + 2 for all sessions). As shown in Table 1, the considered forest therapy sessions were performed with fair weather, without rain, and with comfortable temperatures within the range of 18 to 25 °C.

In Appendix A, Appendix A shows representative pictures of the local environments of the 8 sites listed in Table 1, while Appendix A shows the content of the boards displaying the instructions, in Italian and English languages, read by the participants (SG sessions) or verbally communicated by the therapist (TG sessions).

### 2.2. Outcome Measures

All participants reached each forest therapy site autonomously and, upon arrival, they were required to fill out the informed consent form. Then, every participant was presented with an anonymous questionnaire (with a unique ID code), aimed at collecting information about gender, age, and other personal features. Then, they were asked to fill out two psychometric questionnaires, the State–Trait Anxiety Inventory (STAI) and the Profile of Mood States (POMS) questionnaires.

STAI is a validated questionnaire used to measure state and trait non-disorder-specific anxiety, in both healthy and clinical populations ([53]). It consists of a 20-item self-report measure of anxiety, using a 4-point Likert-type scale for each item. It has two scales: state anxiety, i.e., how one feels at a given moment (STAI-S), and trait anxiety, i.e., how one generally feels (STAI-T), both consisting of 20 items. Total scores were calculated by summing up the items in each respective anxiety subscale (ranging from 20 to 80), where higher scores reflect greater levels of anxiety.

The POMS is a validated questionnaire consisting of 40 items, each measured with a 5-point Likert scale, related to domains such as tension, depression, anger, vigor, fatigue, confusion, and self-esteem ([20]). Both POMS and STAI questionnaires have been widely employed in the environmental psychology research, including recent applications involving healthy cohorts ([28]; [40]) and clinical cohorts ([31]; [50]), as well as in settings comparable to the one examined in this study ([52]).

STAI-S was chosen as the outcome measure of state anxiety, as it has been often used with clinical and subclinical study populations and can detect subtle changes in anxiety levels, even when there are large differences in baseline scores, as occurs with highly heterogeneous participants in forest therapy sessions, showing varying degrees of vulnerability to anxiety ([46]).

The STAI-S and POMS questionnaires were administered immediately before and after each forest therapy session, in order to intercept any short-term effects (changes from baseline) elicited during the sessions. Change-from-baseline STAI-S scores, along with the domain of self-esteem (POMS-esteem) and the total mood disturbance (POMS-TMD), the latter computed as the sum of negative moods (anxiety, depression, anger, fatigue, and confusion) subtracted by positive moods (vigor and self-esteem), were assumed as the main outcome measures. The STAI-T questionnaire was administered only at baseline to better characterize the participants based on their trait anxiety, which reflects the stable long-term tendency toward anxiety, and contributed to the homogeneity analysis of groups across different sessions.

### 2.3. Missing Data

To address missing responses on the questionnaires, we used k-nearest neighbors (KNN) imputation, which estimates missing values by referencing the most similar response patterns from other participants ([3]; [58]). This approach was selected because it maintains multivariate relationships among items without imposing distributional assumptions, making it well suited for Likert-type psychological data. Imputation was performed separately within each TG–SG session pair, to avoid potential bias that could arise from pooling heterogeneous populations across different sites or times. The procedure was limited to cases with up to 20% missing responses per participant on a given questionnaire; cases with more than 20% missing responses were excluded from the specific analysis.

Although more advanced approaches, such as expectation–maximization ([15]), or multiple imputation ([47]), are available, our focus was on ensuring complete datasets for descriptive and comparative purposes rather than for parameter estimation. Moreover, given the modest proportion of missing data and the exploratory nature of this study, we did not conduct a full set of sensitivity analyses with alternative imputation methods.

Calculations were performed in Python (version 3.13) and implemented in the KNNImputer routine of the scikit-learn package ([42]). Responses were rounded to the nearest integer and constrained within the expected Likert ranges (1–4 for STAI items; 0–4 for POMS items).

### 2.4. Homogeneity Across Groups

To evaluate the baseline comparability of the groups, we examined demographic characteristics such as gender and age class using chi-square tests of homogeneity ([34]). In addition to significance testing, we calculated Cramer’s V ([12]) as a measure of association, interpreting effect sizes according to Cohen’s ([11]) benchmarks (0.10 = small, 0.30 = medium, 0.50 = large). This combined approach provided both statistical and substantive assessments of group equivalence.

Contingency tables were constructed to examine the relationship between group membership and categorical demographic variables (i.e., gender and age class). Chi-square tests of homogeneity were performed using the scipy.stats module ([60]), with Fisher’s exact test automatically applied for 2 × 2 tables with low expected frequencies. The strength of associations was quantified using Cramer’s V, computed from chi-square statistics. Data wrangling and tabulation were performed with pandas ([35]).

Homogeneity was further assessed based on STAI-T, i.e., the baseline anxiety levels, reflecting stable individual differences in anxiety proneness that extend beyond demographic factors such as gender and age. Assessing STAI-T distributions across groups allows evaluation of baseline psychological homogeneity, ensuring that differences between the TG and SG interventions were not confounded by underlying trait anxiety ([2]). The use of baseline anxiety, represented by STAI-T, is further supported by the evidence provided by a large-scale study, which found that visiting protected areas improved mental health of initially mentally unhealthy individuals by nearly 2.5 times more compared to mentally healthy individuals ([5]). STAI-T levels were compared across paired TG-SG sessions in the same way as explained in Section 2.5, because STAI-T is homogeneous in nature to STAI-S.

### 2.5. Outcome Analysis

The significance of differences in the mean values of the outcomes considered from the responses to the psychometric questionnaires, i.e., STAI-T, STAI-S, POMS-esteem, and POMS-TMD, was assessed on paired TG–SG sessions within each site using the Wilcoxon–Mann–Whitney non-parametric rank test, which does not assume normal distribution and is widely used in psychological science and particularly in forest healing studies ([31]). The effect size of each intervention was assessed using Cohen’s d index, a metric also widely employed in the environmental psychology field ([36]). Cohen’s d index complements *p*-values by providing meaningful information, especially in studies with limited size samples ([13]). Given the exploratory nature and small per-session sample sizes, *p*-values were not adjusted for multiple comparisons.

Finally, to examine changes in variability across each psychological domain, we employed the Brown–Forsythe test, a robust alternative to Levene’s test that assesses equality of variances based on deviations from the median, and Welch’s test, which adjusts the degrees of freedom to provide unbiased mean comparisons under heteroscedasticity ([14]). Both tests have been successfully used in psychological science ([19]).

The dataset was organized in Microsoft^®^ Excel^®^ for Microsoft 365 MSO (Version 2509, Microsoft, Redmond, WA, USA), and all the statistical tests mentioned in this section were developed and performed within the same software environment.

### 2.6. Illustrative Economic Assessment

To provide an illustrative estimate of the economic value of professional guidance, it was assumed that repeated TG or SG experiences could lead to sustained improvements in emotional functioning. To date, most studies of nature-based interventions have focused on short-term psychological outcomes, with little attention to persistence or economic implications ([6]). Markell and Gladwin found that repeated 4 h SG sessions, conducted once a week for four weeks, were able to maintain the benefits on positive affect and well-being over the following month ([32]). More recent evidence showed that forest- and nature-based interventions can yield sustained improvements in subjective well-being. Chauvenet and collaborators found that a 12-week nature walking program (about 5 h per week) not only produced a higher increase in the Personal Well-Being Index (PWI) at the end of the intervention compared to a control group, both groups consisting of women only, but also produced a longer persistence of the effects, with little decline over three months after the end of the intervention and with leftover benefits lasting as much as 12 months, especially when training in team ([8]). The same study found that mentally unhealthy individuals at baseline showed the greatest leftover benefit 3 months after the program.

Based on the available evidence, the following assumption was made: average per-person changes in STAI-S, POMS-esteem, and POMS-TMD, derived from an annual program consisting in 25 sessions per year, i.e., once a week for four consecutive weeks, followed by a 4-week break, were assumed to translate into equivalent long-term changes in the respective domains, i.e., to the permanent stabilization of the domains considered on new levels. Then, projected changes in mental health domains were converted into gains in QALYs and further into economic value. The method proceeded as follows.

#### 2.6.1. Standardization of Change

For each outcome *j* (either STAI-S, POMS-TMD, or POMS-Esteem), the standardized mean change was computed using Equation (1):(1)$$z_{j}=\frac{∆j}{{SD}_{baseline,j}}$$
where Δ*j* is the difference in the mean between post intervention and baseline, and *SD_baseline_*_,*j*_ is the standard deviation of the outcome *j* at baseline. For STAI-S and POMS-TMD, where reductions indicate improvement, the sign was inverted so that a positive *z_j_* always indicated beneficial change. *SD_baseline_*_,*j*_ was calculated separately for each intervention arm and session, ensuring that standardization reflected the variability inherent to each specific group. This approach aligns with the psychometric tradition of expressing meaningful change in units of baseline variability ([39]).

#### 2.6.2. Mapping to QALY Gain

A half-standard deviation (0.5 × SD) improvement in mental health was associated with an utility gain of approximately +0.07 on the EQ-5D scale, a short generic patient-rated questionnaire for subjectively describing and valuing health-related quality of life ([25]). Accordingly, each standardized change was scaled as shown in Equation (2):(2)$$U_{j}=\left(2\times 0.07\right)z_{j}=0.14z_{j}$$
where *U_j_* is the annualized QALY gain associated with outcome *j*. This function has not been empirically validated for the specific instruments, such as STAI and POMS, or population considered in this study, and should be considered as heuristic.

The inclusion of esteem-related affect as an outcome domain is consistent with recent EQ-5D research demonstrating that self-confidence is a relevant health attribute, conceptually overlapping with quality-of-life valuation and increasingly tested as a bolt-on dimension in patient studies ([45]; [56]).

#### 2.6.3. Accounting for Statistical Significance and Inclusion

Each domain was assigned a weight by a binary flag *f_j_* (1 if the change was statistically significant or explicitly included, 0 otherwise), according to Equation (3):(3)$$U_{j}^{*}=f_{j}U_{j}$$

In particular, with reference to the method explained in Section 2.5, *f_j_* was set to 0 for outcomes showing non-significant changes, preventing non-significant outcomes from contributing spuriously to the economic value.

#### 2.6.4. Multivariate Combination

Because STAI-S, POMS-TMD, and POMS-Esteem are correlated, as POMS-TMD subsumes anxiety and esteem dimensions ([20]; [57]), a multivariate adjustment was applied. The three standardized scores were combined using a Mahalanobis-type distance, as per Equation (4):(4)$$d=\sqrt{z^{T}\sum^{-1}z}$$
where *z* = (*f_STAI-S_z_STAI-S_*, *f_POMS-TMD_z_POMS-TMD_*, *f_POMS-esteem_z_POMS-esteem_*), and Σ is the correlation matrix of the observed changes. To reduce the influence of sampling noise in small groups while avoiding spurious associations across heterogeneous populations, correlations were pooled across matched TG and SG sessions conducted at the same site and period (e.g., S1_TG with S1_SG), but not across different sites or seasons. This ensures that the covariance structure reflects the shared context of each study site while preserving ecological validity.

The overall annual QALY gain was then given by Equation (5):(5)$$U_{total}=0.14d$$

#### 2.6.5. Monetization

The projected QALY gains were monetized using commonly adopted Italian cost-effectiveness thresholds *λ* (EUR 20,000–EUR 50,000 per QALY) ([48]), as per Equation (6):(6)$${Value=U}_{total}\times \lambda$$

The analytic approach used is consistent with recent evidence showing that changes in subjective well-being indices following nature-based interventions can be validly translated into QALY gains and monetized values. In large-scale Australian studies, improvements in PWI from a walking-in-nature therapy program were mapped to QALY changes and valued in monetary terms, confirming both the feasibility and policy relevance of such health-economic translation ([4]; [8]).

#### 2.6.6. Uncertainty Analysis

Uncertainty was quantified by nonparametric bootstrapping with 2000 replicates ([17]). Resampling was conducted within each session file independently, so that the distribution of changes and correlations was preserved separately for each TG or SG group. For each replicate, *U_total_* and its monetized value were recalculated, producing percentile-based 95% confidence intervals. It should be noted that the bootstrapped confidence intervals reflect sampling variability but do not account for model-structure uncertainty. In addition, cost-effectiveness acceptability curves (CEACs) were derived, reporting the probability that *Value* > 0 across conventional willingness-to-pay thresholds ([18]).

## 3. Results and Discussion

### 3.1. Data Imputation

Out of a total of 282 records, and following data imputation, the number of records available for further analysis was as follows: STAI-T, 274 (97.2%); STAI-S before sessions, 260 (92.2%); STAI-S after sessions, 273 (96.8%); POMS-TMD and POMS-esteem before sessions: 274 (97.2%); POMS-TMD and POMS-esteem after sessions: 274 (97.2%). No paired TG-SG sessions retained less than 85% of original records in any domain.

### 3.2. Pairwise Group Homogeneity

Table 3 shows the level of pairwise group homogeneity, assessed based on gender, age, class, and STAI-T of each pair of TG and SG groups. Homogeneity of groups was supported by *p* > 0.05 and effect size (Cramer’s V or Cohen’s d) < 0.5 for all variables; small imbalance of groups was associated with *p* ≤ 0.05 and effect size <0.5, or *p* > 0.05 and effect size ≥0.5 for at least one variable; non-homogeneity was associated with other cases.

Groups in sessions S1_TG and S1_SG, S6_TG and S6_SG, and S8_TG and S8_SG, were homogenous; all the other paired groups showed a small imbalance in one variable, and no paired groups showed non-homogeneity. All sessions were retained for further analysis.

### 3.3. Pairwise Comparison of Effects

Table 4 shows the significance level and the effect size of the changes observed in the psychological domains considered, i.e., STAI-S, POMS-esteem, and POMS-TMD, along with the significance level of variance changes across all paired sessions.

Aside from the similar and comparatively poorer performance in sessions S6_TG and S6_SG, TG sessions generally produced significantly superior outcomes in at least one domain or in most of domains, relatively more frequently in self-esteem and anxiety, both in terms of significance and effect size. The overall comparative superiority of therapist-guided experiences was also observed for sessions S2_TG and S2_SG, which were performed in an urban riverine park. An aggregated view of effect sizes across sites is provided in Appendix A.

The forest therapy trail of sessions S6_TG and S6_SG, which delivered the poorest performance, was characterized by a steep initial climb (95 m of elevation gain in just 300 m of linear route), followed by crossing an evident black pine plantation, both features being susceptible of hindering restoration and healing effects and likely dominant compared to the effects of different guidance models. None of the other trails included steep climbs or plantations.

The increase in positive emotions arising from the exposure to natural environments was ascribed a specific role for subjective well-being ([1]). Earlier, Stellar and colleagues found significant and inverse associations between positive emotions and inflammation markers ([55]), although causality appeared complex and likely bidirectional ([30]). A possible physiological interpretation in the context of the exposure to natural environments was recently provided by Chen and colleagues, based on the consolidated reduction in cortisol in the blood ([10]). Cortisol is a hormone associated with stress and anxiety, leading to immunosuppression and hindrance to the activation of immune natural killer cells; its reduction can enable the activation of natural killer cells via the exposure to plant-emitted monoterpenes, likely mediated by the effect of monoterpenes on the gut microbiota ([10]). Finally, the short-time period of the interventions cannot be considered a hindrance to such a biological pathway, since positive emotions can influence inflammatory markers within just a few hours ([55]). However, the present study did not collect physiological data or monoterpene exposure; these mechanisms are therefore hypothetical in this context and derived entirely from prior literature.

Along with mean-level shifts, significant reductions in variance were occasionally observed only for therapist-led guidance, specifically for the domains STAI-S (session S1_TG, and close to significance for sessions S2_TG and S5_TG) and POMS-TMD (session S3_TG), suggesting a convergence of participants’ responses following a forest therapy intervention. These reductions in variance could be consistent with a regulatory stabilization process and notions of emotional homeostasis ([22]; [38]). However, emotion-regulation mechanisms were not measured directly, and therefore this explanation cannot be confirmed.

As an example, Figure 2 shows the changes in psychological domains for sessions S1_TG and S1_SG, for which the TG and SG groups were homogenous, the outcomes were significantly different, and S1_TG showed a significant reduction in STAI-S variance.

The superior outcomes observed in TG forest therapy compared to SG experiences, generally for mean levels and occasionally for variance, can be interpreted as an effect of delivery mode rather than intervention content. In TG sessions, the therapist’s voice provided social presence and co-regulation, reducing cognitive demands and fostering deeper attentional engagement, whereas the SG participants had to interrupt immersion to read and self-pace written instructions. This difference is consistent with evidence that social cues and reduced cognitive load enhance restoration and flow ([23]; [59]). Therapist-led guidance might also activate expectancy and ritual effects, known to amplify psychological benefits ([43]), and parallels can be drawn with findings from park prescription programs, where structured group activities produced greater stress reduction than individual instructions alone ([44]). Taken together, these mechanisms suggest that guidance delivered interpersonally strengthens the restorative potential of forest immersion beyond what can be achieved through written instruction alone.

### 3.4. Economic Assessment

In a separate, illustrative step, these psychometric changes were translated into QALY-style metrics, which should not be confused with the psychometric findings reported in Section 3.3.

Table 5 shows the mean changes in STAI-S, POMS-esteem, and POMS-TMD and the illustrative economic assessment of the TG and SG guidance modes, computed according to the method described in Section 2.6. The per-person average changes in STAI-S, POMS-esteem, POMS-TMD, and monetization are shown for all sessions, since all paired groups were homogeneous or showed a small non-significant imbalance. Non-significant changes in specific domains were set to 0, and the same domain was excluded from the economic assessment.

The illustrative economic assessment, performed according to the methods presented in Section 2.6, showed remarkably higher annual per person monetization associated with the TG sessions compared to the SG sessions for S1, S2, S3, and S7 and slightly higher for S8; conversely, the SG sessions generated moderately higher monetization for S4 and S5, and slightly higher for S6. All TG sessions generated a significant positive economic value, whereas sessions S3_SG and S7_SG did not show significant monetization. The cross-site median level of annual per person monetization for TG sessions was EUR 4076 at the lower threshold of EUR 20,000/QALY and EUR 10,189 at the upper threshold of EUR 50,000/QALY, while the corresponding levels for SG session were EUR 2462 and EUR 6154, with a ratio of nearly 1.7. Because of the limitations of this economic assessment, these figures and the following ones should be considered as order-of-magnitude estimates.

Appendix A in Appendix A shows the same estimates as in Table 5, limited to the lower threshold of EUR 20,000/QALY and with the progressive inclusion of STAI-S, POMS-TMD, and POMS-esteem. Although these domains showed remarkable mutual correlations, the contribution of domains other than STAI-S contributed substantially to the estimates, up to more than four times compared to the assessment based on STAI-S only, with POMS-esteem contributing more than POMS-TMD and, in the case of sessions S6_TG and S6_SG, being solely responsible for the positive figures.

Based on the experience of the performed forest therapy sessions, a therapist can lead approximately 20 people per session and receive a gross compensation per session of EUR 400, i.e., EUR 10,000 on an annual basis, translating to an annual cost per person of approximately EUR 500. This figure amounts to approximately 12% of the estimated median annual monetization per person of TG sessions at the lower threshold of EUR 20,000/QALY, or an annual saving of approximately EUR 3600 per person (5% and EUR 9700, respectively, at the upper threshold of EUR 50,000/QALY).

The therapist-related cost per person of EUR 500 is approximately 30% of the estimated figure of EUR 1614 (EUR 4076–EUR 2462) for the added annual economic value per person of the TG sessions compared to the SG sessions at the lower threshold of EUR 20,000/QALY and approximately 10% of the corresponding estimated figure of EUR 4035 (EUR 10,189–EUR 6154) at the upper threshold of EUR 50,000/QALY. In other words, a TG-session program would produce an annual per person saving approximately 1.5 times greater than the corresponding figure for a SG-session program (EUR 3576 vs. EUR 2462).

Our findings complement and extend recent work linking forest-based well-being improvements to economic value ([8]). Whereas that study demonstrated the feasibility of mapping PWI gains into QALY increments and AUD values, our approach integrates validated psychometric instruments (STAI, POMS-TMD, and POMS-Esteem) within a comparative TG vs. SG design. Together, these lines of evidence strengthen the case for considering forest therapy not only as an individual health-promoting practice, but also as an intervention with quantifiable economic benefits, suitable for inclusion in healthcare protocols.

### 3.5. Limitations

As stated in Section 2.1, in this retrospective study, participants were not randomly assigned to TG vs. SG sessions; thus, the design is quasi-experimental and susceptible to self-selection bias, and any statistically significant differences between TG and SG should not be interpreted as proof of a causal effect. More structured experiments are needed to confirm the results presented in this study. The generalization of the results is further hindered by the limited number of pairwise sessions and participants and the partial representativeness of the natural environments. Even more important, all experimental sites were located in Italy, which prevents the straightforward extension to other areas and countries, also due to possible cultural differences in forest and nature perception and in the role of therapists. The experiments were performed only with adult volunteers who were sufficiently healthy to participate in a 3 h outdoor program; thus, generalization to children or adolescents, or to clinical populations with severe mental or physical health problems, is not straightforward.

About the handling of missing data, formal sensitivity analyses with alternative imputation strategies were not performed, which might have introduced some residual uncertainty in the estimates. As stated in Section 2.5, another limitation lies in the missing adjustment of *p*-values for multiple comparisons, so the results should be interpreted cautiously, in conjunction with effect sizes. Moreover, despite the hierarchical structure of the data, multilevel models were not used due to few clusters, unbalanced groups and non-normal distributions, choosing instead per-site non-parametric comparisons and then qualitatively synthesizing the pattern across sites. The missing fit of random-effects models could have limited the ability to formally partition variance at the site and session level and, in future work with a larger number of sites and more balanced clusters, multilevel models would be preferable.

Although paired TG-SG sessions were used to minimize systematic differences, residual confounding is likely, such as day-specific conditions (e.g., subtle weather differences, unforeseen disturbances); group composition beyond age and gender (e.g., social dynamics, prior acquaintance); prior affinity with nature or forest environments; and concurrent life stressors not captured by the baseline measures considered. Moreover, the homogeneity assessments shown in Table 3 addressed only a subset of possible confounders (age, gender and STAI-T), and unmeasured factors might partly explain between-session differences.

The economic assessment should be considered illustrative, not validated and likely affected by significant uncertainties due to the oversimplified assumptions. In particular, because of the missing validation of the linear mapping introduced in Section 2.6.2, QALY values should be considered approximate and scenario-based. Moreover, as stated in Section 2.6.6, bootstrap-based confidence intervals reflected sampling variability in a manner consistent with modeling choices, but could not account for model-structure uncertainty.

The absence of longitudinal follow-up constitutes an important limitation, as the persistence of short-term effects was assumed based on limited evidence from other studies; repeated-measures designs with follow-up will be needed to confirm the assumed persistence of short-term effects and, in turn, the design of the annual forest therapy program introduced in Section 2.6.

The comparative estimate of operational savings achieved by means of a TG- compared to a SG-session program did not account for other capital costs, such as equipping and maintaining the signage along the forest immersion trails, which are needed to enable SG sessions and useful for the purpose of attracting further tourism, thus directly benefitting local communities.

The economic assessment was based solely on direct mental health benefits, without accounting either for physiological health or gains in personal productivity. For example, a research in Australia showed that the increase in economic productivity related to independent public visits to protected areas produced a 3 times higher monetization than the reduction in healthcare expenditure ([5]); while mental health improvements are likely associated with an increase in economic productivity and the QALY-based monetization could include at least part of the latter effect, the possibility remains that the figures presented were affected by a systematic underestimation. For example, a recent research based on 1400 interviewees in a survey conducted in Italy found that visiting forests and green spaces at least once in the past year was associated with a higher life satisfaction equivalent to a EUR 11,171 increase in average income ([49]), which is higher than the figures presented in this study even at the upper threshold of EUR 50,000/QALY. Although the life satisfaction approach is completely different from the approach adopted in this study, along with the increase in economic productivity it suggests that the real economic spillover of forest therapy programs might be substantially higher and encompass multiple dimensions. Although specific guidelines were provided to standardize procedures in this study, the effectiveness of TG-type forest therapy interventions might still vary depending on the guide’s interpersonal skills and overall approach.

However, within the limitations of this study, the results concerning both mental health parameters and monetization showed a clear added value of therapist-led guidance, which is therefore not only mandatory for clinically-relevant forest therapy interventions, but also significantly more effective compared with self-guidance.

## 4. Conclusions

The findings of this study suggest that TG forest therapy conveys significantly greater short-term mental health benefits than an equivalent forest immersion (same site and time) conducted in a SG format. The sessions were carried out in eight different forest sites, in natural outdoor settings, with groups of adult participants.

The TG sessions consistently outperformed self-guided ones in reducing state anxiety and total mood disturbance, while simultaneously enhancing self-esteem. These effects were occasionally accompanied by a reduction in variability across specific domains, suggesting a stabilization of participants’ emotional states, in line with theories of psychological homeostasis. Within the limitations of a quasi-experimental setting, these findings underscore that the interpersonal guidance—through the therapist’s voice, presence, and co-regulation—cannot be fully replicated by written instructions, which tend to impose cognitive load that may interrupt immersion and reduce restorative processes.

The illustrative economic analysis further highlighted the added value of therapist guidance, showing that the incremental costs of professional involvement appear modest relative to the order-of-magnitude economic value estimated under the assumptions used in this study. Although the economic results should be meant as a proof-of-concept translation of psychometric changes into QALY-style metrics rather than a formal cost–effectiveness analysis, the findings of this study strengthen the rationale for considering TG forest therapy as a promising candidate for future cost-effectiveness analyses within preventive and complementary health strategies, especially relevant in societies increasingly burdened by rising stress and anxiety linked to urbanization.

Despite all the limitations stated in Section 3.5, the consistency of outcomes across heterogeneous contexts suggests that TG forest therapy deserves broader implementation and systematic evaluation. Future studies should incorporate long-term follow-ups, physiological health indicators, and more refined health-economic models also capable of capturing productivity gains and healthcare savings. Taken together, these results position TG forest therapy as a scalable, low-risk, and economically valuable addition to public health portfolios.

## Figures and Tables

**Figure 1 behavsci-15-01618-f001:**
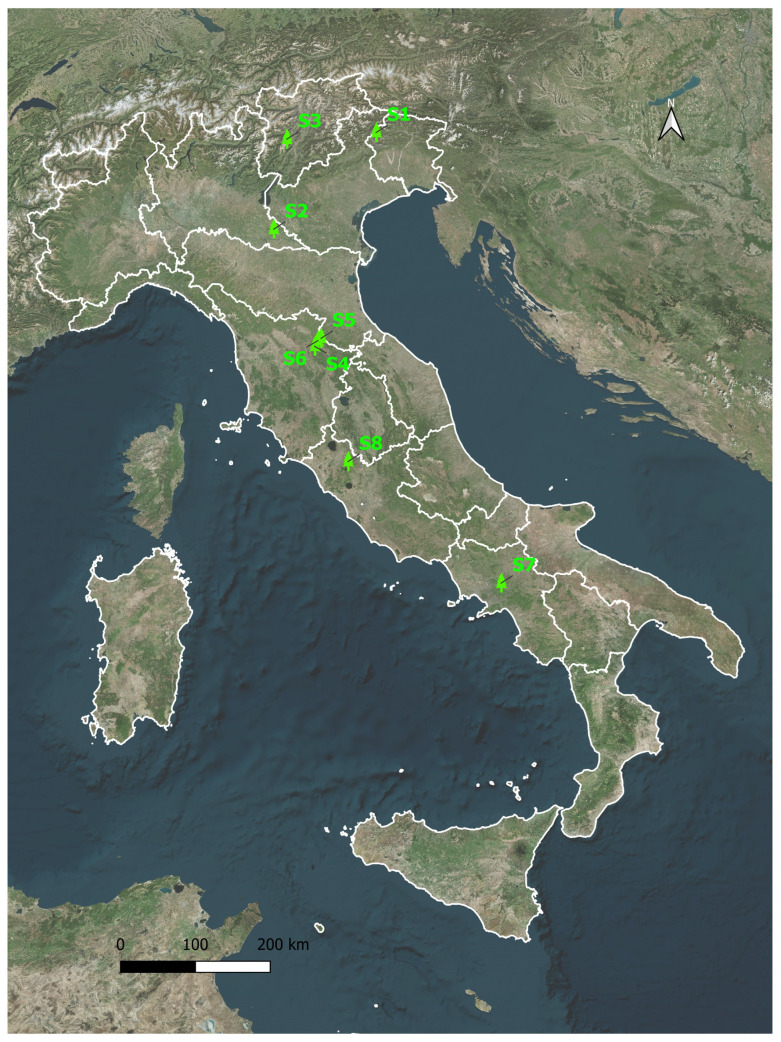
Map showing the distribution of the study sites S1–S8 described in Table 1. The map is plotted in the WGS 84 geographic coordinate system (EPSG:4326).

**Figure 2 behavsci-15-01618-f002:**
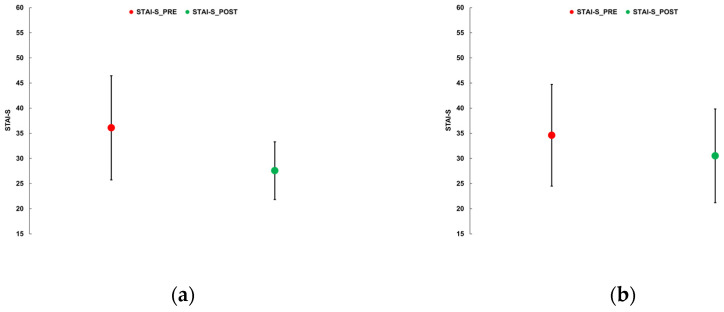
Changes in psychological domains for sessions S1_TG and S1_SG. (**a**) S1_TG, STAI-S; (**b**) S1_SG, STAI-S; (**c**) S1_TG, POMS-esteem; (**d**) S1_SG, POMS-esteem; (**e**) S1_TG, POMS-TMD; (**f**) S1_SG, POMS-TMD. Vertical bars show standard deviation.

**Table 1 behavsci-15-01618-t001:** Geographical, vegetational, and weather data from the TG and SG session sites.

Site Identification		Coordinates ^g^		Site Characteristic			Vegetation	Session	Weather	
ID	Location	Latitude	Longitude	Altitude ^h^	Climb ^i^	Length ^j^	Dominant Forestry Species	Date	T	Cloudiness ^k^
	(Name)	(°N)	(°E)	(m a.s.l.)	(m)	(m)		(dd-mmm-yy)	(°C)	(/8)
S1	Pordenone Refuge ^a^	46.383	12.498	1177	25	2060	Spruce; mountain pine; birch; white willow; alder	4 September 2021	18	1
S2	Mincio Park ^b^	45.168	10.807	18	28	1570	Willow; poplars; oak; turkey oak; alder	18 September 2021	22	3
S3	Tovel Lake ^c^	46.259	10.954	1240	135	2180	Larch; beech; silver fir; sycamore maple; aspen	19 June 2022	23	0
S4	Consuma Pass	43.783	11.588	945	118	1780	Beech; silver fir; Douglas fir; sequoia	13 July 2024	25	0
S5	Borbotto Spring ^d^	43.883	11.688	1242	132	1280	Beech; silver fir	21 July 2024	25	0
S6	Ratoio Hillock ^d^	43.853	11.637	1053	160	1800	Black pine; silver fir; beech	28 July 2024	25	2
S7	Partenio Park ^e^	40.982	14.698	1170	89	1690	Beech	22 June 2025	23	0
S8	Mount Cimino ^f^	42.411	12.204	1010	45	1000	Beech	20 September 2025	23	0

Protected areas: ^a^ Friulian Dolomites Regional Natural Park; ^b^ Mincio Regional Park; ^c^ Adamello Brenta Natural Park; ^d^ Casentino Forests, Mount Falterona and Campigna National Park; ^e^ Partenio Regional Park; ^f^ European Union Natura 2000 Network: Special Area of Conservation and Special Protection Area. Other notes: ^g^ WGS 84 geographic coordinate system (EPSG:4326); ^h^ unit: m a.s.l.; ^i^ total elevation gain (m); ^j^ total length of the trail (m); ^k^ unit: eighths of overcast sky, visually estimated by an expert meteorologist (author F.M.).

**Table 2 behavsci-15-01618-t002:** Participant data from the TG and SG sessions.

Site_Session	Participants							
ID_Type	Gender ^a^			Age Classes (%) ^b^				
	Total; M-F	M (%)	F (%)	≤29	30–44	45–54	55–69	≥70
S1_TG	22; 5, 16	23.8	76.2	4.5	36.4	27.3	27.3	4.5
S1_SG	24; 11, 12	47.8	52.2	8.3	20.8	12.5	41.7	16.7
S2_TG	16; 3,10	23.1	76.9	6.3	0.0	12.5	43.8	37.5
S2_SG	15; 7, 6	53.8	46.2	0.0	33.3	6.7	46.7	13.3
S3_TG	29; 7, 5	19.2	80.8	7.1	7.1	7.1	39.3	39.3
S3_SG	13; 2, 8	20.0	80.0	25.0	0.0	16.7	41.7	16.7
S4_TG	13; 7, 5	58.3	41.7	38.5	7.7	23.1	23.1	7.7
S4_SG	13; 3, 8	27.3	72.7	15.4	7.7	38.5	30.8	7.7
S5_TG	13; 1, 9	10.0	90.0	23.1	23.1	15.4	38.5	0.0
S5_SG	11; 5, 4	55.6	44.4	18.2	9.1	27.3	45.5	0.0
S6_TG	11; 2, 6	25.0	75.0	0.0	54.5	9.1	36.4	0.0
S6_SG	15; 5, 7	41.7	58.3	0.0	53.3	6.7	40.0	0.0
S7_TG	21; 6, 11	35.3	64.7	9.5	19.0	38.1	28.6	4.8
S7_SG	14; 6, 7	46.2	53.8	7.1	0.0	14.3	64.3	14.3
S8_TG	26; 5, 19	20.8	79.2	7.7	34.6	38.5	19.2	0.0
S8_SG	26; 2, 21	8.7	91.3	3.8	23.1	53.8	19.2	0.0

^a^ Total number of participants, number and percentage of males (M) and females (F) limited to available gender declarations; ^b^ limited to available age class declarations.

**Table 3 behavsci-15-01618-t003:** Homogeneity scores across TG and SG groups. * *p*-based significant difference; ^+^ relatively large effect size based on Cramer’s V or Cohen’s d.

Sessions	Gender		Age Class		STAI-T		Assessment ^a^
	*p*	Cramer’s V	*p*	Cramer’s V	*p*	Cohen’s d	
S1_TG S1_SG	0.10	0.25	0.31	0.32	0.43	0.27	H
S2_TG S2_SG	0.11	0.32	0.08	0.52 ^+^	0.75	0.49	SI
S3_TG S3_SG	0.96	0.01	0.60	0.28	0.17	0.60 ^+^	SI
S4_TG S4_SG	0.13	0.31	0.75	0.27	0.05	0.70 ^+^	SI
S5_TG S5_SG	0.03 *	0.49	0.74	0.23	0.98	0.07	SI
S6_TG S6_SG	0.44	0.17	0.96	0.05	0.34	0.36	H
S7_TG S7_SG	0.55	0.11	0.10	0.47	0.11	0.53 ^+^	SI
S8_TG S8_SG	0.24	0.17	0.66	0.18	0.77	0.04	H

^a^ H = homogenous; SI = small imbalance.

**Table 4 behavsci-15-01618-t004:** Level of significance and effect size of changes in mean values and variance in psychological domains. d = Cohen’s d ([36]); *p*_Welch = significance of Brown–Forsythe and Welch tests for variance ([19]). * 0.01 < *p* ≤ 0.05; ** 0.001 < *p* ≤ 0.01; *** *p* ≤ 0.001; # 0.2 < d ≤ 0.5 (small effect size); ## 0.5 < d ≤ 0.8 (medium effect size); ### d > 0.8 (great effect size).

ID	STAI-S			POMS-Esteem			POMS-TMD		
	*p*	d	*p*_Welch	*p*	d	*p*_Welch	*p*	d	*p*_Welch
S1_TG	0.001 ***	1.31 ###	0.008 **	0.001 ***	0.89 ###	0.417	0.004 **	0.77 ##	0.133
S1_SG	0.028 *	0.43 #	0.143	0.061	0.41 #	0.176	0.009 **	0.44 #	0.489
S2_TG	0.001 ***	1.45 ###	0.079	0.001 ***	0.68 ##	0.174	0.062	0.68 ##	0.479
S2_SG	0.037 *	0.45 #	0.432	0.315	0.15	0.386	0.077	0.24 #	0.500
S3_TG	0.001 ***	0.42 #	0.352	0.001 ***	0.89 ###	0.225	0.011 *	0.73 ##	0.048 *
S3_SG	0.409	0.17	0.245	0.215	0.27 #	0.248	0.112	0.29 #	0.446
S4_TG	0.014 *	0.93 ###	0.225	0.043 *	0.78 ##	0.388	0.079	0.45 #	0.481
S4_SG	0.002 **	1.11 ###	0.199	0.067	0.71 ##	0.398	0.079	0.54 ##	0.152
S5_TG	0.005 **	1.49 ###	0.089	0.165	0.42 #	0.200	0.047 *	0.54 ##	0.324
S5_SG	0.012 *	0.73 ##	0.487	0.014 *	0.44 #	0.353	0.035 *	0.74 ##	0.436
S6_TG	0.256	0.28 #	0.396	0.047 *	0.79 ##	0.394	0.245	0.05	0.266
S6_SG	0.062	0.38 #	0.378	0.014 *	0.79 ##	0.084	0.185	0.83 ###	0.134
S7_TG	0.000 ***	1.20 ###	0.391	0.013 **	0.81 ###	0.214	0.134	0.20 #	0.493
S7_SG	0.119	0.31 #	0.328	0.124	0.28 #	0.388	0.147	0.31 #	0.395
S8_TG	0.010 **	0.54 ##	0.470	0.001 ***	0.99 ###	0.405	0.020 *	0.52 ##	0.487
S8_SG	0.000 ***	0.79 ##	0.490	0.016 *	0.60 ##	0.330	0.056	0.44 #	0.467

**Table 5 behavsci-15-01618-t005:** Mean change of STAI-S, POMS-esteem, and POMS-TMD per person, with zero values for non-significant changes, and estimate of annual economic value of TG and SG sessions for the lower and upper thresholds of 20,000 and 50,000 EUR/QALY, respectively. Cross-site medians are also reported.

ID	Mean Changes			EUR per Person, Annual	
	STAI-S	POMS-Esteem	POMS-TMD	EUR 20,000/QALY	EUR 50,000/QALY
S1_TG	−8.5	2.2	−6.9	4967 (95% C.I. 2832 to 8839, pCEAC = 1.0)	12,418 (95% C.I. 7080 to 22,098, pCEAC = 1.0)
S1_SG	−4.1	0.0	−6.3	1537 (95% C.I. 687 to 5118, pCEAC = 1.0)	3843 (95% C.I. 1717 to 12,794, pCEAC = 1.0)
S2_TG	−7.2	2.3	0.0	5313 (95% C.I. 2854 to 12,233, pCEAC = 1.0)	13,283 (95% C.I. 7137 to 30,583, pCEAC = 1.0)
S2_SG	−2.9	0.0	0.0	1167 (95% C.I. 258 to 5034, pCEAC = 1.0)	2917 (95% C.I. 645 to 12,586, pCEAC = 1.0)
S3_TG	−3.5	1.6	−7.4	5438 (95% C.I. 2594 to 9050, pCEAC = 1.0)	13,595 (95% C.I. 6485 to 22,625, pCEAC = 1.0)
S3_SG	0.0	0.0	0.0	0 (95% C.I. 0 to 11,428, pCEAC = 0.343)	0 (95% C.I. 0 to 28,569, pCEAC = 0.343)
S4_TG	−7.8	2.6	0.0	3765 (95% C.I. 1924 to 7607, pCEAC = 1.0)	9413 (95% C.I. 4810 to 19,018, pCEAC = 1.0)
S4_SG	−6.4	0	0	5221 (95% C.I. 2447 to 9468, pCEAC = 1.0)	13,052 (95% C.I. 6118 to 23,670, pCEAC = 1.0)
S5_TG	−8.2	0.0	−5.8	2617 (95% C.I. 2061 to 6472, pCEAC = 1.0)	6544 (95% C.I. 5153 to 16,180, pCEAC = 1.0)
S5_SG	−7.7	1.6	−10.6	4588 (95% C.I. 2487 to 10,307, pCEAC = 1.0)	11,469 (95% C.I. 6217 to 25,767, pCEAC = 1.0)
S6_TG	0.0	1.7	0.0	2909 (95% C.I. 900 to 9225, pCEAC = 0.999)	7274 (95% C.I. 2249 to 23,063, pCEAC = 0.999)
S6_SG	0.0	2.4	0.0	3386 (95% C.I. 1771 to 10,765, pCEAC = 1.0)	8467 (95% C.I. 4427 to 26,913, pCEAC = 1.0)
S7_TG	−6.2	2.4	0.0	3678 (95% C.I. 2168 to 7574, pCEAC = 1.0)	9195 (95% C.I. 5419 to 18,935, pCEAC = 1.0)
S7_SG	0.0	0.0	0.0	0 (95% C.I. 0 to 4605, pCEAC = 0.351)	0 (95% C.I. 0 to 11,512, pCEAC = 0.351)
S8_TG	−4.5	2.5	−6.6	4386 (95% C.I. 2634 to 7071, pCEAC = 1.0)	10,965 (95% C.I. 6584 to 17,676, pCEAC = 1.0)
S8_SG	−5.8	1.6	0.0	3802 (95% C.I. 2409 to 6116, pCEAC = 1.0)	9504 (95% C.I. 6021 to 15,291, pCEAC = 1.0)
Median across TG sessions	−6.7	2.25	−2.9	4076	10,189
Median across SG sessions	−3.5	0	0	2462	6154

## Data Availability

The raw data supporting the conclusions of this article will be made available by the authors on request. Such data will also be openly shared in the form of a structured database in an upcoming study.

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
