# Peer review of "Therapist-Guided Versus Self-Guided Forest Immersion: Comparative Efficacy on Short-Term Mental Health and Economic Value"

_behavsci, 2025, doi:10.3390/bs15121618_

Round 1
Reviewer 1 Report
Comments and Suggestions for Authors
Overall I found this article to have an interesting take on the comparison of therapist-guided versus self-guided forest immersion. The unique take on the economic value between these two methods can both be seen as an altruistic search for financial benefit of forest immersion therapy and also a means to justify this type of therapy (more self-serving). Either way there are benefits.
I did find it interesting that while there was a bit of an anomaly around the S6 data (steeper incline of trail) that could certainly have increased anxiety and negative mood levels, self esteem was impacted in a different way. Leaving room for a potential sense of personal self accomplishment unrelated to whether there was a therapist or not.
I don't have any specific concerns about the methodology or results/conclusions though as mentioned in the limitations the economic valuation does gloss over the complexity of the total economic values.
There are definitely some typos that still need to be worked out of the article. A few specific examples:
- Line 79 - "provide also an added" - remove also
- Line 85 - not a typo but the term "Far Eastern" is a very outdated term
- Line 213 - "patients cohorts" - awkward - maybe clinical cohorts?
- Line 538 - "our the figures" - remove our
- Line 555 - "by a reductions" - should be reduction
Reviewer 2 Report
Comments and Suggestions for Authors
This is a unique paper that compares forest therapy accompanied by a therapist-guide with self-walking. The forest walking paths in Italy are also very beautiful.
This is a unique attempt, but please add your thoughts on the following points:
This study concluded that having a therapist-guide accompanies participants and increases their mental health and economic value.
However, while a therapist-guide may be a necessary condition, it is not yet a sufficient condition.
This is because participants are significantly influenced by the therapist-guide's ability and personality.
A therapist-guide with a careless personality or who has poor interpersonal skills may significantly affect the effectiveness of forest therapy and may also reduce its economic benefits. In fact, there are some therapists and guides who are "not good at interpersonal relationships."
Please add your thoughts on this point.
Reviewer 3 Report
Comments and Suggestions for Authors
The study investigates whether TG forest therapy is superior to SG. The question is clear, and relevant in environmental psychology and preventive health. It applies validated tools, it introduces an economic valuation using QALY-based modeling, bridging psychology and health economics.
The “novelty” is somewhat incremental. Similar studies exist as the authors acknowledge, though none applied QALY monetization. The economic model is conceptually innovative but methodologically fragile due to oversimplified assumptions and absence of validation. Framing short-term limits policy applicability since health economics relies on long-term cost–benefit evidence.
The objectives are explicitly stated, coherent with the analysis. Appropriate use of prepost design and matched comparisons. Limitation section should acknowledge no randomization = potential self-selection bias. No long-term follow-up, so “short-term” effect cannot imply persistence. Reliance on non-experimental assignment weakens causal inference.
Missing data handled with KNN imputation, reasonable, though better justified if sensitivity tests were included.
multiple mentioning using ChatGPT for debugging Python code is unnecessary to mention in a scientific article, as it already specified as Acknowledgments (Line 602).
Correct use of Wilcoxon–Mann–Whitney tests and Cohen’s d for effect sizes.
Variance tests (Brown-Forsythe…) used innovatively to assess emotional regulation. Authors infer emotional regulation from reduced variance this interpretation is interesting but speculative.
No correction for multiple comparisons = risk of inflated Type I error.
Lack of random-effects modeling despite hierarchical data (sessions nested in sites).
QALY conversion is based on highly simplified linear mapping (0.14×z), extrapolated from EQ-5D norms without empirical calibration.
Bootstrapping is appropriate but does not compensate for upstream model assumptions.
The economic model should be reframed as illustrative rather than preliminary quantification.
p-values are sometimes treated as proof of causation.
Lack of visual summaries such as forest plots or aggregated mean changes would clarify trends.
Generalizability limited to Italian forests and healthy adults; cultural factors not discussed.
Discussion sometimes blurs psychological and economic interpretations.
The biological discussion regarding monoterpenes, cortisol… lacks direct data from this study. It should be labeled as contextual literature, not as inferred mechanism.
Limited acknowledgment of confounding variables such as weather, group composition, prior nature affinity….
The claim that TG forest therapy is cost-effective is premature; economic modeling is illustrative.
The manuscript is well-written, formal academic tone and good logical flow. However, some redundancy such as repeated methodological rationale, AI mention…
Materials & Methods section is overly long (11 pages).
Fig. 2 and 3 (environment and instructions) are visually pleasing but could move to supplementary material.
Abbreviation list is clear but some acronyms (FH, SI, NH) appear only once and might not need inclusion.
Consistency wording: psychotherapists vs. therapists vs. clinical professionals.
No aggregated summary table comparing overall TG vs. SG averages across sites.
Lines 165-167 (Ethical): to be deleted (duplicated with lines 594-596).
Line 365: Equation should be (6)
Formatting issues: Units (°C, €/QALY) sometimes italicized, sometimes not.
Round 2
Reviewer 3 Report
Comments and Suggestions for Authors
The authors responded adequately to my comments. I have no further comments.